# The Quality of Life in Surgically Treated Head and Neck Basal Cell Carcinoma Patients: A Comprehensive Review

**DOI:** 10.3390/cancers15030801

**Published:** 2023-01-28

**Authors:** Domantas Stundys, Gintare Ulianskaite, Ieva Stundiene, Jurate Grigaitiene, Ligita Jancoriene

**Affiliations:** Institute of Clinical Medicine, Faculty of Medicine, Vilnius University, LT-01513 Vilnius, Lithuania

**Keywords:** basal cell carcinoma, non-melanoma skin cancer, head and neck, quality of life, surgery

## Abstract

**Simple Summary:**

Basal cell carcinoma is the most common skin cancer with increasing incidences every year. The face is considered to be the most affected body part and surgery the most often applied treatment method. Although this tumor rarely metastasizes and is generally considered to cause low morbidity, the established oncologic diagnosis and the existing impairment in facial area as well as the surgical treatment, which may leave postoperative scars and facial disfigurement of varied extent, affect the patient’s quality of life. The aim of this article is to review and summarize current literature on the impact of craniofacial basal cell carcinoma surgical treatment on patients’ quality of life and to compare it before and after the surgery. After conducting a comprehensive review, we conclude that there is a lack of studies assessing the impact of surgical treatment on quality of life exclusively in patients with head and neck basal cell carcinoma.

**Abstract:**

In this review, we examine current literature analyzing the impact of surgical treatment on the QoL in patients with head and neck BCC. A comprehensive literature review was performed using the main databases. As many as six out of 322 articles were selected for the final analysis. The selected articles were published in the period between 2004 and 2021, most published within the last two years. All analyzed studies were prospective. Five out of six studies evaluated NMSC consisting of both BCC and SCC, and only one study selectively evaluated the impact of surgical treatment on QoL in patients with craniofacial BCC. Authors of the selected studies reported that QoL improves following the surgery; however, the effect on QoL varies. Patients’ age, gender, marital status, education level, and employment status had a stronger correlation with QoL postoperatively, especially during the late follow-up period. Younger patients were more bothered by appearance-related issues. One study concluded that elderly patients did not experience a statistically significant improvement in QoL. This literature review demonstrated that there is no clear consensus on the use of a single disease-specific QoL measurement tool. Furthermore, there is a lack of studies assessing the impact of surgical treatment on QoL exclusively in patients with head and neck BCC and studies analyzing the multivariate correlation between QoL and tumor type, size, anatomic site, and treatment outcomes.

## 1. Introduction

Basal cell carcinoma (BCC) is the most common type of non-melanoma skin cancer (NMSC) and one of the most common cancers in the white population [1,2,3]. The highest incidence rates have been reported in Australia (mean incidence of 1000/100,000 inhabitants), the USA (mean incidence of 212–407/100,000 inhabitants), and Europe (mean incidence of 76.21/100,000 inhabitants per year) [4]. The incidence of the disease is increasing significantly each year worldwide, by 4–8% in the USA [1] and 2% in Australia [5]. The disease is most common in the elderly, with the risk of developing BCC being 100 times higher in people aged 55–75 years than in those aged 20 years [5]. However, there is a worrying trend towards a rapidly increasing incidence of BCC in the age group of population under 40 years, especially in women [6].

The most important risk factors for the development of BCC are skin exposure to UV, Fitzpatrick skin types I and II, immunosuppression, exposure to arsenic, ionizing radiation, and genetic syndromes [1,2,5,7,8,9,10]. Around 70–80% of BCC occur on the sun-damaged areas of the face, head, and neck, and less often on the extremities [3,4]. Even though the chances of metastases of BCC are very low, this type of skin cancer could locally invade the surrounding tissues.

The aim of treatment of BCC is to remove the tumor, preserve the function of the affected area, and maintain a good aesthetic appearance. There are many options for both conservative and surgical treatments of BCC. The choice of the method is mainly deter-mined by the risk of tumor recurrence, which is further divided into low-risk and high-risk BCC. Tumor criteria (location, size, histological findings, assessment of tumor margins, possible perineural spread, recurrent tumor) and patient criteria (age, immunosuppression, genetic syndromes, chronic scarring, ulceration, foci of inflammation, history of other malignancies) are considered [7].

The main treatment of BCC is surgical, which is usually chosen according to the risk of recurrence [4,7,9]. If a high risk of BCC recurrence is suspected, a more aggressive treatment approach is selected accordingly. Less aggressive BCC can be treated with minimally invasive surgical (electro destruction, curettage, cryotherapy) or non-surgical methods (radiotherapy, local immunotherapy, photodynamic therapy) [4,7].

Although BCC is usually not life-threatening, the disease affects the largest organ of the body, the skin, which is the most visually prominent and visible to oneself and the others. As the face, head, and neck are most affected, it can significantly affect person’s body image, self-esteem, and quality of life (QoL) [11,12]. The impact on QoL may be due to the skin tumor itself, as well as to the treatment administered, such as symptoms, functional limitations, changes in aesthetic appearance, and additional considerations such as the cost of treatment and interference with activities of daily life [13].

Since the number of BCC patients is increasing every year, and surgery is still considered the main treatment method, more people must deal with emotional and physical consequences of the disease. The aim of this article is to review and summarize current literature on the impact of craniofacial BCC surgical treatment on patients’ QoL.

## 2. Materials and Methods

### 2.1. Search Strategies

A scientific literature search was performed in PubMed, Cochrane, and Web of Science databases from inception to 1 September 2022. The keywords used for the search were groups of words “basal cell carcinoma”, “non-melanoma skin cancer”, and “NMSC” combined with the word “quality of life”. Results were limited to the English language. In total, 322 articles were identified through database searches.

### 2.2. Study Selection Criteria

The chosen articles were evaluated against eligibility criteria:Age: the study included patients over 18 years oldLocation of tumor: craniofacial BCCIntervention: surgical treatment of the face and neck BCCOutcome: QoL was assessed before and after the surgical treatmentArticle types: prospective studies were included

As many as 322 articles were identified through literature database search using the above-mentioned keywords. After a thorough abstract review, 268 articles were excluded. One article was excluded since there was no possibility to retrieve a free full text article. Following exclusion criteria, 47 articles were excluded after a full text review. Six articles were chosen for a final analysis (Figure 1).

Each of the 6 included papers was analyzed by two authors (DS, GU). The main data from the articles was collected and described in the table including authors, study type, description of study, sample size, description of the sample, tools used to evaluate QoL, and main conclusions. The findings were presented chronologically and brought into a review.

## 3. Results

The selected articles were published in the period between 2004 and 2021; most were published in the last two years. All analyzed studies were prospective. QoL indicators were assessed at various time points with the first assessment being before surgery and follow-up assessments carried out at the period from 1 week after surgery up to 5 years. Five out of six studies evaluated NMSC consisting of both BCC and squamous cell carcinoma (SCC), and only one study selectively evaluated the impact of surgical treatment of only craniofacial BCC on QoL. Various questionnaires such as Medical Outcomes Study 36-Item Short-Form Health Survey (SF-36), Visual Analogue Scale (VAS), Functional Assessment of Cancer Therapy—General (FACT-G), the Rosenberg Self-Esteem Scale/UNIFESP-EPM, Dermatology Life Quality Index (DLQI), Skin Cancer Index (SCI), and FACE-Q were used to assess the QoL. A detailed summary of the six selected articles is presented in Table 1.

Rhee et al., in his longitudinal prospective study of 121 NMSC of the head and neck patients, assessed QoL, smoking habits, and sun-protective behavior before and after the surgical treatment. SF-36 and FACT-G were used as QoL evaluation tools. Authors observed only slight changes in QoL measures. The postoperative scar following the treatment was less bothersome than the lesion itself. The location and size of the tumor was not associated with QoL in this cohort of patients. However mental (SF-36) and emotional (FACT-G) domains of QoL showed statistically significant changes. Notably, study participants younger than 65 years and employed demonstrated improvements in emotional and mental health and well-being following the treatment of NMSC, especially between the surgical treatment and the 1-month postoperative visit (*p* < 0.04). In addition, many of those included in the study were more likely to use sunscreen or protective clothing or limit their outdoor presence during peak UV light exposure following the treatment (*p* = 0.001). However, no change in smoking habits was observed. Authors also expressed their doubt in general QoL instruments not being able to capture the specific QoL issues in patients with NMSC. They also presented an idea to develop a disease specific QoL instruments and carry out an additional study in order to investigate anxiety, distress, and disease management strategies in such patients [14].

Another prospective study by Maciel et al. evaluated the QoL and self-esteem in patients with head and/or neck skin cancers. Fifty patients between the ages of 30 and 75 years were enrolled in the study, and their QoL and self-esteem were evaluated preoperatively and 5 years following the surgery. QoL was assessed using the Brazilian version of SF-36, and self-esteem was evaluated using the Rosenberg Self-Esteem Scale. Patients with lesions less than 1 cm in diameter were excluded. Authors observed improvement in mental health (*p* = 0.011) and self-esteem (*p* = 0.002) in patients who underwent surgical treatment in the five-year postoperative period. However, there were no significant differences in relation to other domains of the SF-36 or the self-esteem scale. Five-year follow up resulted in a considerable loss to follow-up of the patients (56%). Authors believe this was due to an adequate treatment performed, the minimally invasive nature of skin cancer, and a less bothersome attitude of patients not willing to return for follow-up visits [15].

A prospective study by Çetinarslan et al. evaluated 255 patients with facial NMSC using the Turkish version of DLQI preoperatively and 3 months after the surgical treatment. Authors also collected and analyzed data on demographic factors that could presumably affect the QoL of the patients, such as gender, educational level, duration of the disease, type of skin lesion, affected anatomical area, and primary or recurrent tumor. At baseline, the most affected subscale was symptoms and feelings in both BCC and SCC groups (*p* < 0.001). The least affected subscale was work and school in patients with BCC. In patients with SCC, treatment was the least affected subscale preoperatively (*p* < 0.001). In regard to tumor localization, the worst lesion site was auricular and preauricular localization. Following 3 months after the surgery, authors reported a significant improvement in QoL in both BCC (3.96 ± 5.14) and SCC (4.49 ± 5.24) patients after the surgery (*p* < 0.001), when compared with the baseline DLQI scores (6.37 ± 6.28 in BCC and 6.35 ± 6.16 in SCC group, respectively). There was no significant difference observed between the QoL of male and female patients both preoperatively and postoperatively, mainly due to DLQI lacking the domain-capturing aesthetic outcomes. Authors also reported the worst DLQI scores in patients with university degrees, due to increased awareness of the disease, and in those with the graft reconstruction, due to increased risk of complications and worse cosmetic results at both the baseline and 3 months postoperatively [16].

García-Montero et al. carried out a prospective cohort study including 229 patients with cervicofacial NMSC patients. A Spanish version of the SCI questionnaire was used to evaluate QoL at the time patients received a diagnosis of NMSC and subsequently at 1 week, 1 month, and 6 months postoperatively. Authors observed statistically significant (*p* < 0.05) differences between the mean scores of the SCI (both overall and for each of the subscales) at the time of diagnosis and at 6 months after surgery. In the overall SCI scores, statistically significant differences were observed by gender (*p* = 0.047), educational level (*p* = 0.019), tumor type (*p* = 0.044), treatment type (*p* = 0.042), and VAS score (*p* = 0.014). The social-aesthetic scale revealed statistically significant changes in gender (*p* = 0.01), marital status (*p* = 0.012), and history of depression and/or anxiety (*p* = 0.002) parameters. Meanwhile, the emotional scale educational level (*p* = 0.002), tumor type (*p* = 0.027), treatment type (*p* = 0.018), and VAS score (*p* = 0.011) demonstrated statistically significant differences. Authors noted that women in this cohort of patients experienced greater improvement in aesthetic appearance domain. However, this may be due to the fact that women may pay more attention to issues related to facial attractiveness than men or are capable of masking the facial imperfections with make-up. Those with primary education reported a higher degree of improvement in the emotional domain; meanwhile, married patients presented the greatest improvement in the social-aesthetic domain. As one of the study limitations, authors admitted the need to perform a larger study and include patients undergoing non-surgical interventions, as this type of treatment is becoming more and more attention in the management of NMSC [17].

A prospective study by Kinde et al. included 45 consecutive periocular NMSC patients who underwent MOHS micrographic surgery and reconstruction. SCI and FACE-Q questionnaires were given to patients preoperatively and subsequently at 1 week and 3 months as QoL measurement tools. Specific demographic and clinical characteristics, which presumably could have influenced patients’ QoL, were also collected (gender, smoking status, history of skin cancer, tumor location and type, and reconstruction method). The study revealed that the total SCI score and all three of the subscales were significantly higher after surgical treatment than preoperatively, especially at the 3-month follow-up evaluation. The FACE-Q scale also demonstrated similar findings. Using SCI scores, older age was associated with improved QoL, while for tumor location at medial canthus and reconstruction by a myocutaneous flap or full-thickness skin graft, female gender and history of NMSC were the predictors of reduced QoL. Authors also reported that higher baseline QoL scores predicted higher postoperative QoL. Limited ethnic diversity and sample size of the cohort as well as a rather short follow-up were the main limitations of the study [18].

A prospective observational study by Sanz Aranda et al. included patients older than 85 years with histologically confirmed head and neck BCC who were asked to answer the Spanish version of the SF-36 before and 3 months after the surgery. 25 patients filled in both preoperative and postoperative SF-36 questionnaires. Authors reported that the only significant change observed in the study cohort was for physical role (*p* = 0.026), and it became worse. Physical role subscale evaluation decreased significantly in patients with multiple comorbidities: physical role and mental health in those with facial BCC, and general health and social function in those with a tumor larger than 1 cm. Based on study findings, elderly patients who underwent surgery for BCC did not experience a statistically significant improvement in QoL. Therefore, surgical treatment of BCC, as a first line treatment, should be thoroughly discussed with patients and relatives, and alternative treatment options should be provided. However, the authors admit that their results should be interpreted with caution due to existing comorbidities in this cohort of patients and their advanced age [19].

As states the Constitution of The World Health Organization, health is described as “a state of complete physical, mental, and social well-being and not merely the absence of disease or infirmity” [20]. QoL is highly dependent on the beforementioned health state since it is highly influenced by physical, mental, and social aspects of well-being.

Most of the reviewed articles demonstrate that BCC mainly affects patients’ mental health, which usually is ameliorated after surgical treatment. Rhee et al. confirmed that changes in mental and emotional domains of QoL are statistically important, especially in the younger-than-65-years-old, employed patient group. The impact on mental health was also proved by Maciel et. al., emphasizing the improvement of patients’ self-esteem. Furthermore, Çetinarslan et al. showed that all DLQI subscales improved after surgical BCC treatment, with symptoms and feelings being the most affected subscale.

In regards to social wellbeing, García-Montero et al. claimed that emotional and social-aesthetic subscales of QoL questionnaires improve 6 months after surgery with the exception of divorced patients and a history of recurrent tumors in emotional subscale. The difference in the social-aesthetic group was noted between sex and people with a history of depression/anxiety with a higher post-treatment improvement in women and people with a history of depression/anxiety. In comparison, Kinde et al. reported an increase in all three subscales of QoL evaluated by SCI 3 months after surgery.

On the other hand, Sanz Aranda et al. presented slightly different study findings in a senior patient group. They were the only authors to reveal that the physical subscale of QoL, mental and social health, decreased after surgery in BCC patients older than 85 years.

A formal meta-analysis was not performed due to the differences in the methodologies of the analyzed studies, various QoL measurement tools used by the authors, different follow-up times after surgery, and rather heterogenous study groups. However, we used forest plots to illustrate the effect sizes of the different studies on the change in QoL after surgery, expressed as standardized mean difference with 95% confidence interval (SMD with 95% CI). One study is not depicted in the plots due to insufficient data.

The effect sizes on the change in QoL are presented in Figure 2, Figure 3 and Figure 4.

## 4. Discussion

The analysis of the selected articles demonstrated that there is no clear consensus on the use of a single disease-specific QoL measurement tool that would allow assessment of the impact of surgical treatment on QoL in patients with BCC. As various instruments are used, parallel comparison of the results obtained within different studies is seldom possible. Moreover, existing QoL measurement instruments are not suited to distinguish and evaluate the impact on QoL specifically in patients with BCC after surgical treatment. However, even if general QoL measurement tools may not be able to assess the specific QoL issues in patients with NMSC, a particular advantage of these instruments is that the scores can be compared with other diseases [14].

The most commonly used questionnaires were 36-SF, FACT-G, DLQI, and SCI. SF-36 was validated in 1992 and is widely used in different medical spheres to evaluate the impact of the disease as well as the different treatment methods on the QoL [21,22,23]. This questionnaire in greater detail evaluates eight different domains: physical functioning, physical role, bodily pain, general health, vitality, social functioning, emotional role, and mental health. However, it is not disease specific. FACT-G was developed and validated in 1993 to evaluate health-related QoL in oncology clinical trials and nowadays is commonly used in cancer patients undergoing treatment. It consists of four main domains evaluating physical, social-family, emotional, and functional aspects that may influence patients’ QoL [24]. DLQI was validated and specifically created in 1994 to evaluate the QoL in patients with skin disease. This questionnaire consists of 10 questions that cover symptoms and feelings, daily activities, leisure, work and school, personal relationships, and treatment [25]. SCI is an NMSC-specific questionnaire that was developed to evaluate the impact of skin cancer on QoL. The questionnaire was validated in 2004 and is considered to be face valid as well. It consists of 15 questions that are related to emotional, social, and appearance subscales [11].

All the above-mentioned instruments evaluate the QoL, but they differ in their range of application. The most versatile and applicable to various kinds of health conditions is SF-36. Although such a wide application provides universality, the questionnaire does not allow a specific assessment of the impact of skin cancer on the QoL. Furthermore, the FACT-G questionnaire is specific to oncological diseases, but it is not adapted to measure the impact of skin cancer, which differs from other oncological diseases. Another questionnaire widely known in dermatology is the DLQI, which is specific to skin diseases but is more suitable to be used in patients suffering from chronic skin diseases than in those with skin cancer. The most recent of all described questionnaires is the SCI, which is the most specific of all the aforementioned tools, as it assesses the impact on QoL of skin oncological diseases.

Chernyshov et al. in 2019 acknowledged the problem of QoL measurement in skin cancer patients and stated the position of the European Academy of Dermatology and Venereology Task Forces. After conducting research on the most used QoL assessment tools, they have stated that EORTC QLQ-C30, the Functional Assessment of Cancer Therapy-Melanoma (FACT-M), SCI, SF-36, and the DLQI were the most-used questionnaires. The task forces recommend the use of cancer-specific questionnaires in the late stages of cancer (EORTC QLQ-C30) and more skin-cancer-specific questionnaires: the melanoma-specific FACT-M and skin-cancer-specific SCI questionnaires. According to the task forces, the other questionnaires are not currently recommended in this context [26].

## 5. Conclusions

The authors of aforementioned studies reported that following the surgical treatment the patients with cervicofacial NMSC experience an improvement in QoL, especially in regards to their emotional status and mental health. Interestingly, this QoL improvement varies from slight to significant. Patients’ age, gender, marital status, education level, and employment status were those variables that had a stronger correlation with QoL postoperatively, especially during the late follow-up period. Younger patients were more bothered by appearance-related issues, such as postoperative scars, facial disfigurement, and self-image. However, elderly patients reported a negative association of surgical treatment and their QoL, as their general health, physical role, mental health, and social subscale evaluation decreased postoperatively, based on comorbidities, tumor site, and diameter, respectively.

This literature review demonstrated that there is a lack of studies assessing the impact of surgical treatment on QoL exclusively in patients with head and neck BCC. Most reviewed studies collectively include NMSC, which include SCC, BCC, and sometimes also include actinic keratosis and Bowen’s disease. Although BCC represents most of the sample in the selected studies, even a small number of other types of NMSC may have an influence on the results of the study, due to the tangible differences in the investigation and treatment tactics of each of these diseases, e.g., extensive surgical modalities, recommended safety margins, probability of recurrence, and/or metastases.

It was also observed that there are no conducted studies that evaluate the association of specific tumor type, size, anatomic site, cure type, and treatment outcomes with the change occurred in the QoL and self-esteem in patients with head and neck BCC, both in terms of the conspicuous nature of the disease, its rapidly increasing incidence rates worldwide, and appearance changes related to surgical treatment.

## Figures and Tables

**Figure 1 cancers-15-00801-f001:**
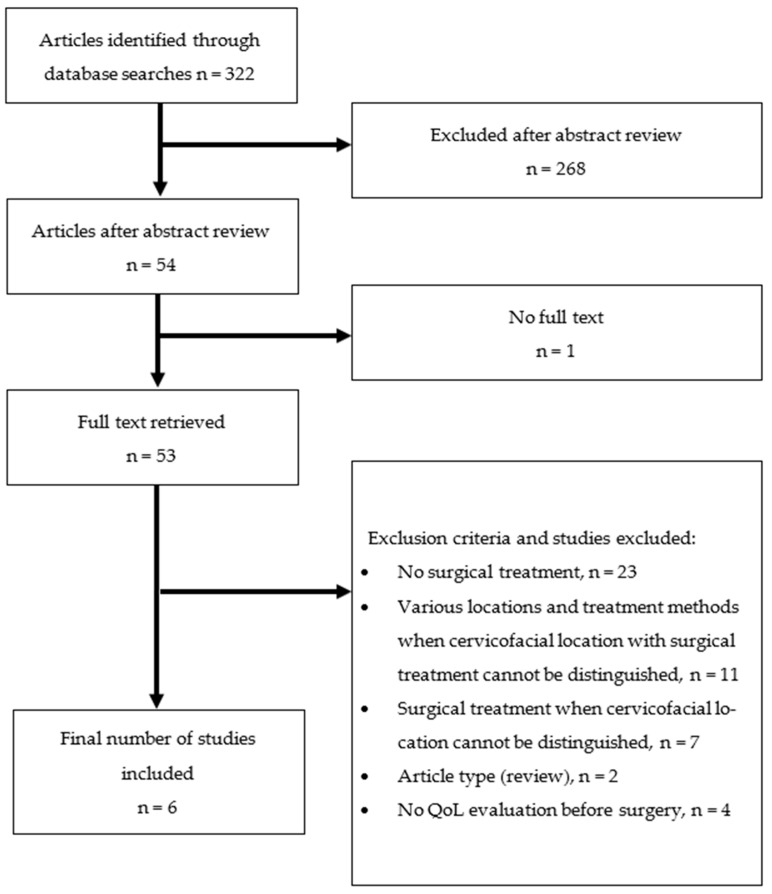
Article selection flow diagram.

**Figure 2 cancers-15-00801-f002:**
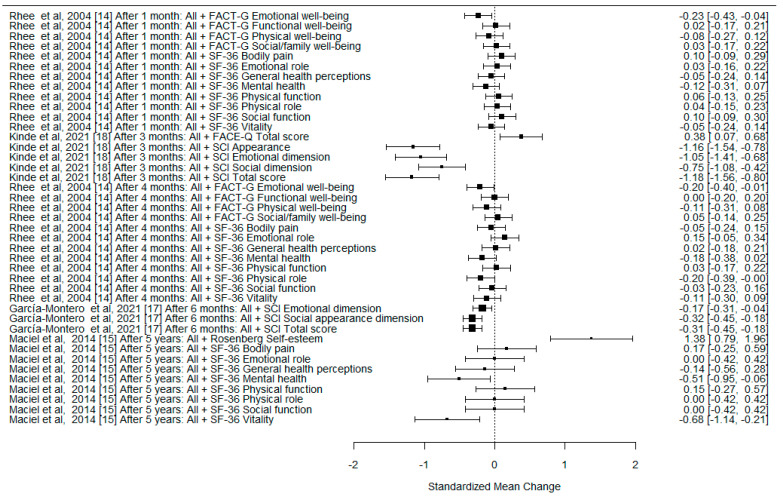
The forest plot for differences in the preoperative and postoperative QoL measures for the overall group (not stratified by histological type). SMD with 95% CI.

**Figure 3 cancers-15-00801-f003:**
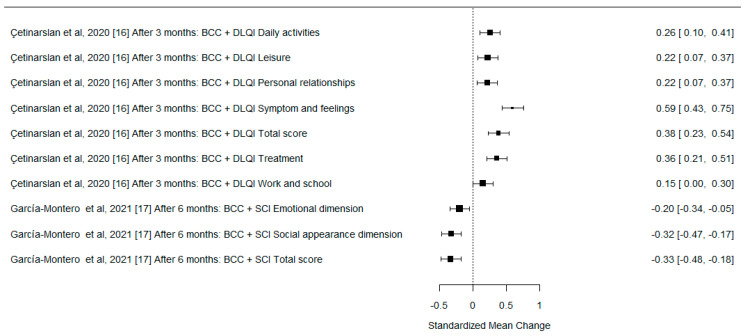
The forest plot for differences in the preoperative and postoperative QoL measures for the BCC group. SMD with 95% CI.

**Figure 4 cancers-15-00801-f004:**
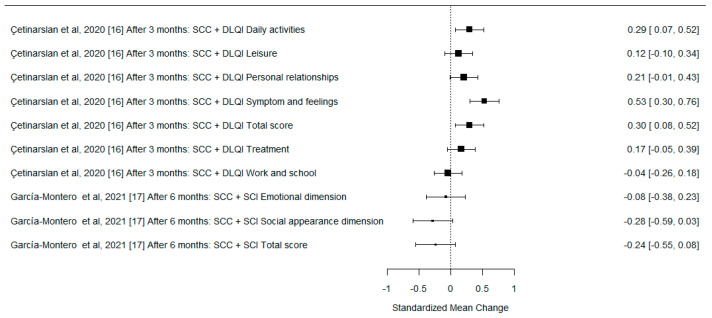
The forest plot for differences in the preoperative and postoperative QoL measures for the SCC group. SMD with 95% CI.

**Table 1 cancers-15-00801-t001:** Detailed summary of the selected articles.

Author	Study Type	Study Description	Sample	Sample Description	QoL Outcome Measures	Main Conclusions
Rhee et al., 2004 [14]	Longitudinal prospective research	QoL assessment of cervicofacial skin cancer treated with Mohs surgery. QoL data collected at baseline, 1 month and 4 months	n = 121—initial visit n = 105—first follow up n = 102—second follow up	Biopsy-proved NMSC cervicofacial skin cancer: n = 103—BCC, n = 16—SCC, n = 2—other	SF-36, 10-cm visual analog scale (VAS), Functional Assessment of Cancer Therapy-General (FACT-G)	Little change of QoL was noticed following the treatment of NMSC; the improvements in emotional, and mental health following treatment of NMSC were established (specifically <65 years and employed patients).
Maciel et al., 2014 [15]	Prospective, analytical clinical study	Assessment of late impact of surgical treatment of cervicofacial skin carcinomas on QoL and self-esteem. QoL data collected at baseline and 5 years after surgery	n = 55—initial visit n = 22—5-years after surgery follow up	Biopsy-proved NMSC cervicofacial skin cancer larger than 1 cm: n = 19—BCC, n = 3—SCC	SF-36, the Rosenberg Self-Esteem Scale/UNIFESP-EPM	Improvement in mental health and self-esteem was observed in the late postoperative period after surgical treatment of NMSC.
Çetinarslan et al., 2020 [16]	Prospective study	Determination of the factors affecting QoL and the effect of surgical treatment on QoL of patients with facial NMSC. QoL data collected at baseline and 3 months after surgery	n = 255—initial visit	Histologically or clinically diagnosed facial BCC or SCC: n = 174—BCC, n = 81—SCC	DLQI	The QoL is minimally affected in patients with NMSC using DLQI; the QoL 3 months after surgery showed a significant improvement in patients with facial NMSC.
García-Montero et al., 2021 [17]	Prospective cohort study	Identification of the factors related to the favorable evaluation of QoL during follow-up after treatment of cervicofacial NMSC. QoL data collected at the time of diagnosis, 7 days, 1 month and 6 months after treatment	n = 229—initially included n = 220—completed questionnaires	Cervicofacial NMSC, confirmed by skin biopsy: n = 179—BCC, n= 41—SCC Type of treatment: n = 190—surgery n = 19—photodynamic therapy n = 3 imiquimod 5% n = 8 cryotherapy or electrosurgery	SCI, VAS, clinical interview	Scores of the SCI improve after the treatment of cervicofacial NMSC.
Kinde et al., 2021 [18]	Prospective study	Measurement of QoL of individuals with surgically treated periocular NMSC. QoL data collected at baseline, 1 week and 3 months after surgery	n = 57—enrolled patients n = 45 completed questionnaires	Patients diagnosed for the first time with periocular NMSC who underwent Mohs micrographic surgery and reconstruction: n= 37—BCC n= 8—SCC	SCI, FACE-Q	Mohs resection of periocular NMSC patients demonstrated reduced QoL as measured by the SCI and FACE-Q surveys; the significant improvement of Qol after this surgery was reported. The highest improvements were in the late postoperative period.
Sanz Aranda et al., 2021 [19]	Prospective observational study	QoL in histologically confirmed BCC patients older than 85 years treated with surgery. QoL data collected before and 3 months after surgery	n = 48—presurgery questionnaire n = 25—postoperative survey	Histologically confirmed BCC patients older than 85 years	Spanish SF-36	A significant improvement of QoL after surgery was not detected; the authors believe that surgery as a first-line treatment for BCC should be discussed with patients and their caregivers or relatives, along with alternative options.

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
