# Peer review of "The Quality of Life in Surgically Treated Head and Neck Basal Cell Carcinoma Patients: A Comprehensive Review"

_cancers, 2023, doi:10.3390/cancers15030801_

Round 1
Reviewer 1 Report
This review features discussions on the quality of life (QoL) in surgically treated basal cell carcinoma in head and neck. Basal cell carcinoma is the most common type of non-melanoma skin cancer. The primary treatment of basal cell carcinoma is surgical, usually chosen according to the risk of recurrence. The surgical treatment of the face, head, and neck can significantly affect a person's body image, self-esteem, and QoL. The authors reviewed and summarized articles on the impact of surgical treatment of craniofacial basal cell carcinoma on a patient's QoL. As a result, the authors argued that there is a lack of research evaluating the impact of surgical treatment on QoL for patients with basal cell carcinoma of the head and neck. The theme set is interesting, but there are some problems with the structure of this review. The following points should be clarified.
In the Results, the authors only listed the summaries of several manuscripts. I recommend that the authors set some themes and summarize articles according to the theme.
As this manuscript is a comprehensive review, the authors should summarize the treatment and mechanism of basal cell carcinoma in head and neck.
Author Response
Dear Reviewer,
Thank you for your time while reviewing our manuscript. We very much appreciate your insight. Taking into account your comments we have included several additional paragraphs to our manuscript.
As we don’t analyze the BCC treatment methods per se, we have included a short summarizing description of the general idea about the basal cell carcinoma treatment – main first line treatment methods, treatment goals as well as some major concerns to be taken into account while planning the treatment process in the head and neck area.
We have also introduced some themed summary of the result section, as you recommended. For themed grouping we decided to go with the health definition presented in the Constitution of WHO in 1946, mainly grouping the findings of the reviewed studies based on “physical“, „mental“ and „social well-being“ issues.
Once again thank you for your review.
Sincerely,
Domantas Stundys - on behalf of all the authors
Reviewer 2 Report
The reviewed manuscript is dealing with an estimation of therapeutic efficacy of head and neck basal cell carcinoma. The first finding was a discovery of lack of interest in the disease attributed to head and neck region. Wrong classification? Head and neck surgeons and oncologists do not bother about non-melanoma skin cancer. An incidence of NMSC indicates first on Australia with strongly operated sun attracting inhabitants to spending time on a beach. I am aware of social campaign to inform people about negative effects of overusing sun-burning. It looks it was not followed by appropriate studies done by finally … Lithuanians.
The study is estimated two aspects: mental and physical health. Because of a few papers only accessible for analysis final conclusion are somehow obscure with a stress for positive aspect of mental health. Further, estimation of QoL was dome at very different time periods extended from one week to 5 years. Anyway the authors have analyzed QoL attentively and sufficiently. Hence, the study is worth publishing.
Minor remark: Citation 18 seems to do not have sufficient bibliographic data.
Author Response
Dear Reviewer,
Thank you for your time while reviewing our manuscript. We very much appreciate your insights on the complexity of the discussed issue. I fully agree with you about the observed perspective of head and neck surgeons towards the NMSC cancer in general. There is a trend in attitude towards BCC being a “mild” form of cancer. However, our team in everyday practice meets facial BCC patients who are overly worried about the diagnosis, treatment process, visual result of the treatment outcomes and more. This beyond all doubts reflects on the QoL of the patients, even if the worse thing in their lives at that particular moment is some “mild” cancer. We are glad you found our manuscript scientifically sound, rather well-structured and worth publishing.
We have updated the Reference 18, thanks to your attentive reviewing.
Once again thank you for your time.
Sincerely,
Domantas Stundys - on behalf of all the authors
Reviewer 3 Report
This fairly comprehensive review of the literature highlights the flaws in capturing meaningful data on evaluating the impact of surgery on QoL in BCC patients.
Whilst there are differences in the type of data collected in the different studies there are similarities across some if not all the studies which should be amenable to some attempt at statistical analysis, and if this would included then I think it would increase the impact of the review and make it more suitable for publishing in Cancers.
Author Response
Dear Reviewer,
Thank you for your time while reviewing our manuscript. We very much appreciate your comment on including some type of statistical analysis. Based on your remark we have extracted and aggregated some less heterogenous data presented in the reviewed studies. As the study methodologies, QoL measurement tools, follow-up intervals rather differ and similar data sets are not equally present in the reviewed studies a formal meta-analysis deemed impossible to be performed. However, we managed to calculate the effect size for overall, BCC and SCC group respectively. The forest plot diagrams were created by R metaphor package and included in the manuscript as Figures 2-4.
Once again thank you for your valuable remark.
Sincerely,
Domantas Stundys - on behalf of all the authors
Reviewer 4 Report
Dear authors, The paper is well done and I haven't question or correction to do.
Author Response
Dear Reviewer,
Thank you for taking your time to review our manuscript. We appreciate your comments as an encouragement to further continue our research in this field.
Sincerely,
Domantas Stundys - on behalf of all the authors
Round 2
Reviewer 1 Report
The authors have addressed the points which I noted.